# Identifying Conservation Introduction Sites for Endangered Birds through the Integration of Lidar-Based Habitat Suitability Models and Population Viability Analyses

Erica Marie Gallerani [1,2,*], Lucas Berio Fortini [3], Christopher C. Warren [4] and Eben H. Paxton [3]

1 Department of Geography, University of California Los Angeles, 1255 Bunche Hall, P.O. Box 951524, Los Angeles, CA 90095, USA
2 Hawai'i Cooperative Studies Unit, University of Hawai'i at Hilo, 200 W. Kawili St., Hilo, HI 96720, USA
3 Pacific Island Ecosystems Research Center, U.S. Geological Survey, P.O. Box 44, Hawai'i National Park, HI 96718, USA; lfortini@usgs.gov (L.B.F.); epaxton@usgs.gov (E.H.P.)
4 Haleakalā National Park, National Park Service, P.O. Box 369, Makawao, HI 96768, USA; christopher_warren@nps.gov
* Correspondence: ericagallerani5@g.ucla.edu

**Abstract:** Similar to other single-island endemic Hawaiian honeycreepers, the critically endangered 'ākohekohe (*Palmeria dolei*) is threatened by climate-driven disease spread. To avert the imminent risk of extinction, managers are considering novel measures, including the conservation introduction (CI) of 'ākohekohe from Maui to higher elevation habitats on the Island of Hawai'i. This study integrated lidar-based habitat suitability models (LHSMs) and population viability analyses (PVAs) to assess five candidate sites currently considered by managers for CI. We first developed an LHSM for the species' native range on Maui. We then projected habitat suitability across candidate CI sites, using forest structure and topography metrics standardized across sensor types. Given the structural variability observed within the five candidate sites, we identified clusters of contiguous, highly suitable habitat as potential release sites. We then determined how many adult individuals could be supported by each cluster based on adult home range estimates. To determine which clusters could house the minimum number of 'ākohekohe birds necessary for a stable or increasing future population, we conducted PVAs under multiple scenarios of bird releases. We found that canopy height and relative height 90 had the greatest effects on model performance, possibly reflecting 'ākohekohe's preference for taller canopies. We found that a small release of at least nine pairs of equal sex ratios were sufficient for an 80% chance of success and a <1% chance of extirpation in 20 years, resulting in a minimum release area of 4.5 ha in size. We integrated the results of the LHSM and PVA into an interactive web application that allowed managers to consider the caveats and uncertainties associated with both LHSMs and PVAs in their decision-making process. As climate change continues to threaten species worldwide, this research demonstrates the value of lidar remote sensing combined with species-specific models to enable rapid, quantitative assessments that can inform the increasing consideration of time-sensitive conservation introductions.

**Keywords:** 'ākohekohe; lidar; sensor fusion; population viability analysis; habitat suitability models; conservation introductions; texture analysis

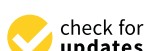



## 1. Introduction

The escalating global impacts of climate change, anthropogenic development, disease outbreaks, and other dynamic factors have pushed many species to the brink of extinction, rendering their historical habitats and ranges unfit to sustain viable populations [1]. Increasingly, managers are considering relocating plants and animals outside their historical range to locations deemed more suitable for species survival. These actions, termed conservation introductions (CIs), have received widespread attention in recent years as traditional

conservation approaches fail to address current threats [2]. However, a CI is a leap into the unknown, introducing species to ecosystems where they did not evolve, potentially leading to numerous unforeseen consequences [3]. This inherent uncertainty stresses the importance of careful planning in release site selection, as habitat suitability in these novel environments will greatly affect the likelihood of success [2].

Hawaiian honeycreepers are a group of species in which CIs are being considered to prevent extinction [4,5]. Avian malaria (*Plasmodium relictum*) is the primary driver of recent declines in multiple honeycreeper species, placing them at risk of imminent extinction. Avian malaria is vectored by the southern house mosquito (*Culex quinquefasciatus*), and both the mosquito and the malaria parasite require sufficient temperatures to develop [6]. This has historically kept high-elevation forests free of disease because temperatures were too cool for mosquito populations to develop. However, climate change has facilitated the rapid spread of *C. quinquefasciatus* into higher elevations on the Hawaiian Islands, threatening disease-sensitive honeycreeper species that had once been safe from disease [4,7–9]. Isotherms associated with critical temperatures for *Plasmodium* development (13 °C) have historically delineated disease lines on the Hawaiian islands in both current and future climatic conditions [10]. 'Ākohekohe (*Palmeria dolei*) is a federally endangered honeycreeper species endemic to the islands of Maui and Moloka'i [11], now restricted to a range of less than 3000 ha above ~1600 m of elevation in east Maui [12]. The nectar of 'ōhi'a (*Metrosideros polymorpha*) flowers comprises 50–75% of their diet [13]. Post-natal juveniles often travel long distances in search of food resources, which can include lower elevations, placing them at an elevated risk of contracting malaria [14]. From 2001 to 2017, the 'ākohekohe range contracted by 61%, with population estimates declining by 78% to a current population of 1768 (95% confidence interval = 1193–2411) birds [12]. Expert assessment of current declines and remaining population size indicated that 'ākohekohe could become extinct within 10 years [5].

Given the current and projected population trends for 'ākohekohe, novel approaches to managing the species are being considered, such as possible inter-island CIs [5]. Given the narrow elevational extent of 'ākohekohe's habitat on Maui, the potential CI of the species to the Island of Hawai'i, which has more native habitat within suitable elevations for forest birds, could help safeguard the species from future warming-induced extinction risk [8,15]. A previous strengths, weaknesses, opportunities, and threats (SWOT) assessment started the process of evaluating the plausibility of this action by comparing the quality of candidate CI sites on the Island of Hawai'i for the release of kiwikiu (*Pseudonestor xanthophrys*), 'ākohekohe, 'akikiki (*Oreomystis bairdi*), and 'akeke'e (*Loxops caeruleirostris*) [16,17]. This analysis identified several candidate CI areas for 'ākohekohe on the Island of Hawai'i (Figure 1). These sites were evaluated qualitatively based on the input of expert biologists, which provided an important first step in the process of identifying quality release sites. However, a more quantitative assessment of strength, weakness, and threat indicators could better inform the viability of the considered sites. An additional limitation of this qualitative approach was that each site was viewed as a whole rather than considering how the indicators varied across the landscape within each site.

A major challenge in achieving successful CI arises from an incompatible habitat at the release site compared with the habitat of origin [18,19]. A global review of CI success found that habitat-related factors were the largest drivers of initial declines for birds and other taxa released in novel habitats [2]. In fact, expert biologists in Hawai'i gave some of the highest weights to species-specific habitat indicators during a previous SWOT analysis. Therefore, in our research, we sought to elucidate this issue of habitat compatibility by trying to identify suitable habitats for 'ākohekohe based on forest structure and topography within the candidate CI areas. The particular diets of 'ākohekohe as nectarivores indicate a specialized use of forest habitat, making them particularly vulnerable to forest cover and connectivity [13,20,21]. We chose to derive forest structure and topographic metrics from various light detection and ranging (lidar) point clouds that matched those used to build lidar-based habitat suitability models (LHSMs) for Hawaiian honeycreepers in previous

research [22,23]. Gallerani et al. (2023) [22] used maximum entropy modeling (Maxent) to transfer habitat suitability models of 'akikiki and 'akeke'e from Kaua'i to east Maui. The 'akikiki and 'akeke'e models performed exceedingly well (AUC > 0.80), showing that a similar methodology is reasonable for the implementation of Maui birds on the Island of Hawai'i.

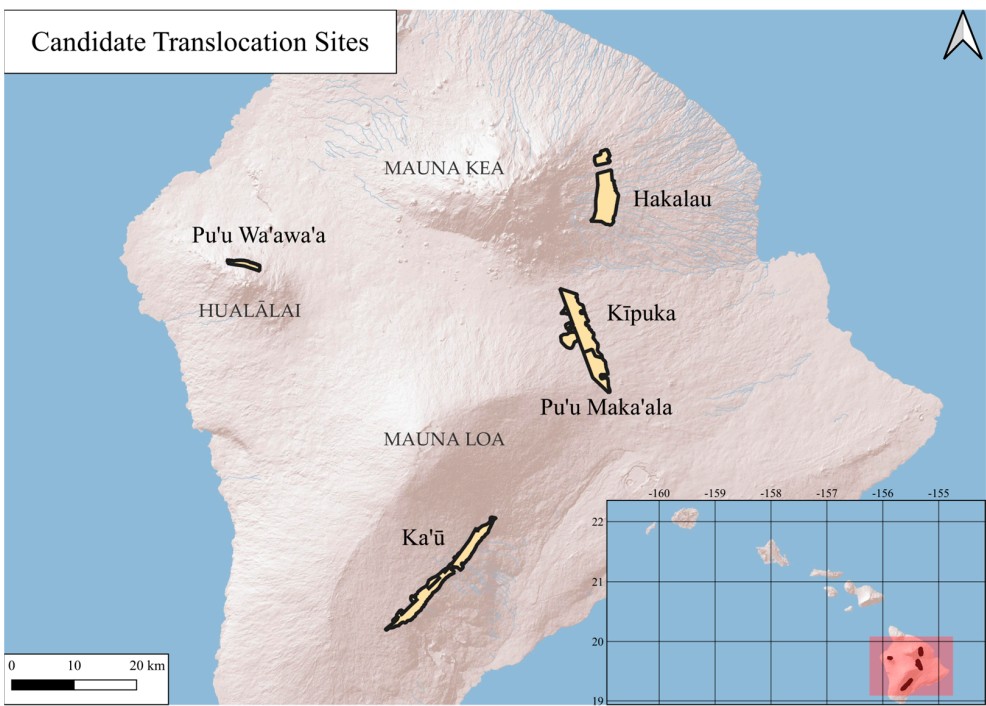

**Figure 1.** Candidate sites on the Island of Hawai'i that were investigated through a 2022 SWOT analysis for conservation introductions of 'ākohekohe and further analyzed by this study.

While habitat-related factors pose an important challenge to successful CIs, quantifying habitat suitability alone does not guarantee long-term population viability. Ensuring that translocated populations can grow and persist over time also requires identifying areas of contiguous suitable habitat of sufficient extent. Therefore, in our work, we enacted the novel approach of incorporating habitat suitability models with population viability analyses (PVAs) to better inform decisions on optimal release sites. By integrating habitat quality and quantity constraints, managers can choose among potential sites with greater confidence in supporting self-sustaining 'ākohekohe populations into the future. We hypothesize that our analysis will fill knowledge gaps from previous qualitative and quantitative CI assessments for Hawaiian forest birds through an integrated approach of remote sensing modeling and demographic modeling. In addition, this analysis will help identify essential areas of connected suitable habitat for 'ākohekohe release within each of the top candidate sites. This study aimed to build the necessary evidence base for a potentially critical conservation action to save not only 'ākohekohe, but other endangered Hawaiian forest birds.

## 2. Materials and Methods

### 2.1. Study Area

We collected species and remote sensing information from the island of Maui, where 'ākohekohe are currently found, to develop LHSMs for the species. The wet, windward slopes of Haleakalā on Maui are home to six Hawaiian honeycreeper species, including 'ākohekohe, of which two are endangered species and another is a threatened species. This area consists of federal land, private land, and several state-managed forest reserves and Natural Area Reserves (NARS) and is collectively referred to as east Maui throughout this paper. East Maui contains some of the most pristine remaining native forests on

Maui [24–26]. These forests are characterized as dense, montane wet forests with an average rainfall across the ʻākohekohe habitat of ~4.7 m annually [27].

The Island of Hawaiʻi is the youngest, largest, and tallest island of the archipelago, and consequently is where all current candidate CI sites for the species are being considered. We considered the five candidate CI sites identified in the previous SWOT analysis, all located within state and federal lands above the critical isotherm for *Plasmodium* development (1675 m in elevation) [17]. Hakalau National Wildlife Refuge is owned by the U.S. Fish and Wildlife Service (USFWS) and is located on the windward side of Mauna Kea. The Kaʻū site consists of a state Forest Reserve and Hawaiʻi Volcanoes National Park Kahuku Unit on the southeastern slopes of Mauna Loa. The remaining sites, Puʻu Waʻawaʻa, Kīpuka, and Puʻu Makaʻala, are all managed by the Hawaiʻi State Department of Land and Natural and Resources (DLNR). Kīpuka is located in the Upper Waiākea Forest Reserve and Kīpuka Ainahou Nēnē Sanctuary. Puʻu Makaʻala NAR is located just south of Kīpuka, and is also one of the National Science Foundation's (NSF) National Ecological Observatory Network sites (NEON). All of these sites receive an average of ~2.2 m of annual rainfall [27]. Finally, Puʻu Waʻawaʻa is a Forest Bird Sanctuary located on the northern slopes of Hualālai volcano on the leeward side of the island. Being on the leeward side of the Island of Hawaiʻi, this site receives less annual rainfall, with an average of 0.7 m [24].

*2.2. Lidar and Bird Data Acquisition*

ʻĀkohekohe occurrence locations in east Maui were collected during the Hawaiʻi Forest Bird Surveys, point count surveys conducted at regular intervals by experienced observers from multiple agencies [24,28]. From 2012 to 2018, these surveys yielded 587 occurrence locations of individual ʻākohekohe. These occurrences were concentrated in The Nature Conservancy's Waikamoi Preserve, Hanawī NAR, and Haleakalā National Park. In order to avoid sample selection bias in the spatially clustered ʻākohekohe occurrence data, we implemented background weight correction with the FactorBiasOut algorithm, as described by Gallerani et al. (2023) [22].

Mapping forest structure over large distances, such as Maui to the Island of Hawaiʻi, usually involves heterogeneous datasets of small footprint lidar data with varying acquisition parameters [29]. The Global Airborne Observatory (GAO) collected high-resolution aerial lidar data over east Maui using an Optech HA 500 dual-channel sensor in January 2018. The lidar coverage area contains high-value bird habitat and some lower elevation forests. High-resolution aerial lidar point clouds were obtained for all candidate CI sites on the Island of Hawaiʻi from several different sources. The U.S. Geological Survey (USGS) provided lidar data for parts of Puʻu Makaʻala, Hakalau, and Kīpuka sites and was the sole source of lidar data for Kaʻū and Puʻu Waʻawaʻa sites [30]. USFWS also provided lidar data for the Hakalau site [31]. Additional coverage of the Puʻu Makaʻala site was obtained via the NSF [32]. USGS data were acquired using a Leica single-photon lidar (spl) sensor, whereas the NSF data were acquired using Optech Incorporated Airborne Laser Terrain Mapper (ALTM) Gemini sensors as the full-waveform lidar instrument. USFWS data were acquired using a Riegl LMS-Q680i. Relevant specifications of the sensors were compared (Table A1) to determine which sensor would collect point clouds most similarly to the GAO Maui data. We matched all data from the Island of Hawaiʻi to the GAO data to ensure the transferability of the LHSMs built on the Maui lidar data. The Optech Gemini sensor used in the NSF collections was found to have the most similar beam divergence (0.25 mrad), pulse repetition frequency (~100 kHz), and wavelength (1064 mm) to the Optech HA-500 sensor used in the east Maui collection. We ran regression models for each vegetation structure metric between NSF and USGS data in the Puʻu Makaʻala and Kīpuka sites, and between USGS and USFWS at the Hakalau site. The areas of overlap were inspected to make sure that they covered a variety of forest structure types so that they could be representative of lidar collected at all sites. These regression relationships were then used to sensor-correct all USGS metrics to more closely resemble data collected by the NSF sensor. A summary of the regression results is presented in Table A2. Additionally,

the USFWS Hakalau data were adjusted in a two-step process based on its relationship to the USGS data.

### 2.3. Lidar Data Processing

All lidar point clouds were processed using the lidR package in R (version 4.3.1) [33]. The raw lidar point cloud data were first preprocessed to remove noise and outliers. Points with height values below 0 m were dropped to remove erroneous data below ground level. Then, statistical noise filtering was applied to remove points with height values more than 1.2 times the 95th percentile height within each 100 m cell. This effectively removed spuriously high points not representative of true vegetation structures. After denoising, a ground classification algorithm (multiscale curvature classification) was applied to the data, using the default scale parameter of 1.5 and curvature threshold of 0.3, as they yielded satisfactory ground classification results [34]. Once ground classification was complete, we applied a nearest-neighbor interpolation algorithm to convert absolute point heights to heights relative to the ground surface. For each non-ground point, the height value was calculated relative to the interpolated ground surface below it. After height normalization, additional outliers with heights exceeding 50 m or below $-50$ m were removed to eliminate any residual noise. This processing chain produced a cleaned, normalized point cloud ready for further analysis.

Forest structure metrics, including a canopy height model and four relative height metrics (RH25, RH50, RH75, and RH90), were derived at a 100 m resolution using standard pixel metrics in the lidR package. The canopy height model, referred to simply as canopy height throughout, is the average height of vegetation returns, and not the top of canopy. Relative height metrics describe the forest structure by representing the height distribution of a point cloud. The RH metrics selected here shed light on the vertical profile of the forests in the candidate CI sites. For example, a higher RH25 value means a taller understory and a large spread between RH25 and RH75; RH90 indicates a complex forest with a diverse range of height structures. We wrote a custom function to derive a canopy density metric using 1.37 m as the height threshold to match Maui data. Canopy density represents the percentage of lidar returns above a certain height threshold. The 1.37 m threshold represents the height at which diameter at breast height (DBH) is measured for individual trees. Terrain metrics (i.e., the slope and topographic wetness index) were derived from the 1/3 arc-second (~10 m) bare-earth digital terrain model (DTM) available through the USGS 3D Elevation Program (3DEP). Slope was derived using the raster package in R [35]. Topographic wetness index (TWI) represents the potential for water accumulation based on topographic characteristics such as the slope and contributing area, and has implications for the presence of mosquito breeding habitats [23]. This measurement is typically used to assess the extent of the upslope contributing area that flows through a particular pixel in the DTM weighted by the tangent of the slope. We calculated TWI using a Python script from Fricker et al. (2015) used in ArcGIS Pro (version 3.1.0) which can be found here: https://github.com/africker/Topographic-Wetness-Index (accessed on 30 June 2023) [36]. Once we derived the terrain variables, we clipped and resampled them to match the vegetation structure metrics derived at the 100 m scale in the lidR package.

### 2.4. Habitat Suitability Models

We conducted a novelty analysis using the command prompt tool Novel [37] to analyze the potential transferability of Maui lidar-based models to the Island of Hawai'i. This analysis found that slope most often contributed to the novelty of the Island of Hawai'i pixels when compared with the east Maui lidar data. Furthermore, we plotted the distributions of the nine metrics on east Maui with distributions of the same variables from the candidate CI areas on the Island of Hawai'i (Figure 2). These histograms show that the lidar coverage on east Maui has a wide range of slope values, centered around 25 degrees, whereas the candidate CI sites on the Island of Hawai'i have a narrower distribution, centered around 5 degrees (Figure 2d). Similarly, the elevational range on

east Maui is much larger than in the candidate CI sites, with the Island of Hawai'i sites having a higher median than east Maui (Figure 2a). When transferring models to novel habitat, collinearity is of particular concern as the relationships between variables may differ from island to island [38]. To determine collinearity, we performed Spearman rank correlation between the nine lidar-derived metrics on the Island of Hawai'i (Table A3). Given the collinearity concern and the fact that mismatched distributions from the training site (Maui) to the projection site (Island of Hawai'i) can reduce transferability, we chose to exclude variables with the most novel distributions on the Island of Hawai'i that were highly correlated to other variables that had more similar distributions. With the high levels of correlation between variables being excluded and those being retained to build the model, not much information is lost to the model from the exclusion of said variable. Elevation stands as an exception in our decision-making process. The large discrepancy in elevation distribution between Maui and the Island of Hawai'i would likely reduce model transferability between islands. Additionally, because elevation can be a strong predictor of forest bird distributions due to avian malaria [8], the preliminary model runs showed a prevalent elevational pattern that obscured the importance of other site-specific factors. The primary focus of our work was to identify compatible areas for Cis based on forest structure characteristics, as important determinants of forage and nest habitat [22]; therefore, we dropped elevation as a variable in the model. The finalized list of variables used in the models is shown in Table 1. The metrics include three of the four structural type classes defined by a recent review of lidar vegetation metrics in bird species distribution modeling, cover, height, and vertical variability [39]. Bakx et al. (2019) found that metrics related to canopy height and canopy cover were those that most often led to an effective explanation of species distributions. Although vertical variability metrics, in this case RH25 and RH90, were less effective than cover and height, they were the second most widely used metric class in avian species distribution modeling.

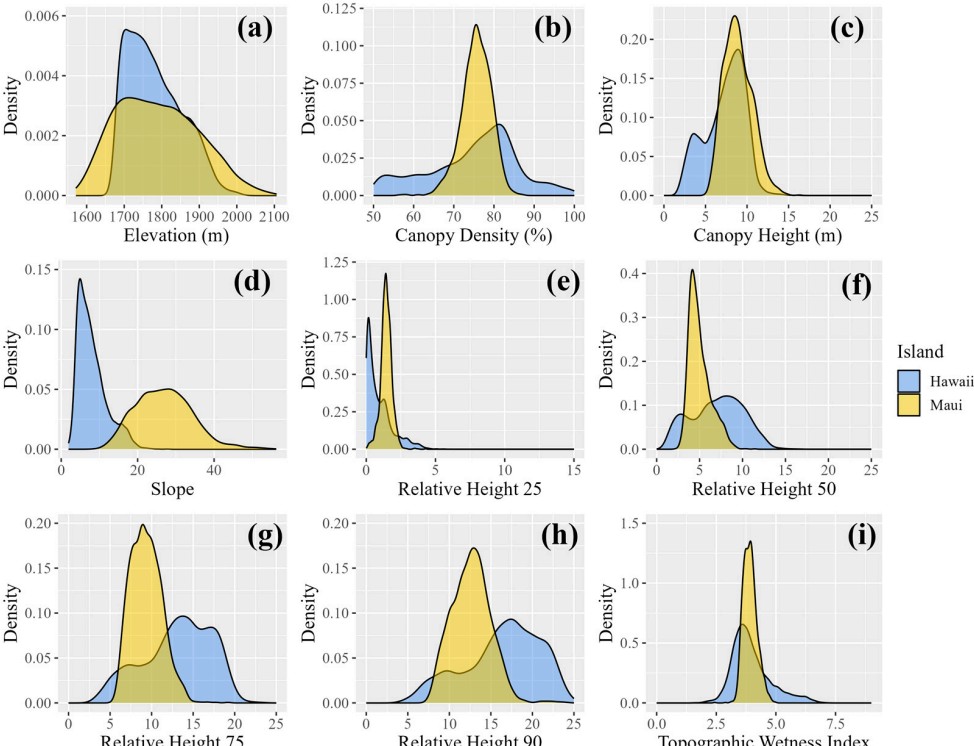

**Figure 2.** Density distributions of the nine lidar-derived metrics with east Maui lidar coverage represented in yellow and the candidate conservation introduction areas on the Island of Hawai'i. The metrics are as follows: elevation (**a**), canopy density (**b**), canopy height (**c**), slope (**d**), relative height 25 (**e**), relative height 50 (**f**), relative height 75 (**g**), relative height 90 (**h**), topographic wetness index (**i**).

**Table 1.** Lidar metrics used as inputs into the 'ākohekohe models.

| Variable Name | Description |
| --- | --- |
| Canopy Density (DNS) | Percentage of lidar points > 1.37 m (4.5 ft) above ground in the selected region |
| Canopy Height (CHM) | Height (m) of the modeled canopy surface above the ground |
| $RH_{25}$ | 25th relative height above the ground for all lidar points in the selected region |
| $RH_{90}$ | 90th relative height above the ground for all lidar points in the selected region |
| Topographic Wetness Index (TWI) | Relative wetness of the selected region within the landscape |

We ran the 'ākohekohe models using the same Maxent software (version 3.3.4 k) following Gallerani et al. (2023) [22]. Maxent is a presence-only machine learning software that models the predicted suitability of gridded metrics based on known locations of individuals. To reduce overfitting, we implemented a bootstrapped resampling method that produces 100 sub-models, averaged to determine habitat suitability. The iterative modeling process can also help overcome the general consensus that finer scales are more appropriate for modeling rare species [40]. We used a moderately fine scale of 100 m for the lidar metrics in order to optimize transferability of these models to novel environments. Area under the receiver operating characteristic curve (AUC) scores, used here to measure model performance, reflect a model's ability to rank observations higher than random background pixels in presence-only modeling [41]. Standard deviations between outputs are used to measure uncertainty. The percentage contribution and permutation importance of each environmental variable are reported to determine which metrics were the most valuable in predicting 'ākohekohe suitability. Percentage contribution is derived from the addition or subtraction of the increase in regularized gain during each iteration of the training algorithm to or from the contribution of a given variable. In the estimation of permutation importance, each environmental variable's values within the training presence and background data are randomly transformed one at a time. The model is then re-evaluated using these permuted data and the resulting reduction in training AUC is normalized to a percentage. Additionally, response curves representing 'ākohekohe suitability models created using only one variable were interpreted to determine species-specific habitat preferences for highly correlated variables.

*2.5. Candidate Translocation Introduction Site Assessment*

We evaluated each candidate CI site by calculating mean 'ākohekohe habitat suitability and uncertainty. The site with the highest mean habitat suitability and lowest mean uncertainty would therefore be considered the most suitable on the grounds of vegetation structure and topography. Some of the candidate CI areas contained lava flows, immature forest, or other land cover types that do not constitute potential habitats for 'ākohekohe. To not bias the summary statistics of these areas, we decided to further segment the 'ākohekohe LHSM results into forested areas only. We used the Food and Agriculture Organization of the United Nations (FAO) definition of forests as follows: "Land spanning more than 0.5 hectares with trees higher than 5 m and a canopy cover of more than 10 percent, or trees able to reach these thresholds in situ" [42]. The mean suitability and uncertainty were then calculated based on strictly forested areas. Another concern for the determination of candidate release sites is the connectedness of the suitable habitat. We therefore used a gray-level co-occurrence matrix (GLCM) with a 3 × 3 moving window to calculate several texture statistics for the LHSM results at each candidate release area. A GLCM in this instance captures the spatial relationship of habitat suitability by analyzing the frequency of pairs of suitability levels at various distances and directions within the 3 × 3 window [43]. We ultimately decided on homogeneity as our main texture statistic, as it quantifies the uniformity of a given image region, providing us with a more comprehensive interpretation

of the candidate release areas. We performed quantile analysis of the homogeneity texture statistic for the 'ākohekohe suitability results to determine an appropriate threshold for homogenous areas. We used the median as our cut-off, equivalent to a homogeneity value of ~0.2. We refer to these areas as homogenous forested areas (HFAs) because they are defined by both forest cover and homogeneity values. We then used the minimum training presence (MTP) threshold defined by Gallerani et al. (2023) [22] to determine which homogenous areas are also suitable for 'ākohekohe release. The areas of clusters that are both homogenous and suitable were then summarized across candidate CI sites.

### 2.6. Population Viability Analysis

To estimate the minimum area and associated number of birds required for a CI release, we performed PVA using stochastic simulations in R. The model was developed considering the available demographic parameters for the species based on previous research. Simulations focused on determining the number of bird pairs needed per release, assuming an equal sex ratio in the releases. Several demographic parameters were integrated into the discrete-time model, including pairs renesting within season, nest success, chicks per nest, juvenile survival rate, and adult survival rate for first year post release (to account for release effects) and long term. A full explanation of these parameters and their values is presented in Appendix B. To account for the inherent stochasticity and uncertainty in the demographic parameters, 10,000 stochastic simulations were executed for each release scenario considered, each generating a unique population trajectory. The stochasticity was incorporated by allowing the adult and juvenile survival rates and nest success to vary randomly within specified bounds derived from the available literature. Further explanation of the stochastic simulations is detailed in Appendix B. A successful run was defined as a simulation where the population was stable or increasing over the 20-year simulation (i.e., the population size at year 20 was equal to or greater than the initial population size). The proportion of successful runs was then calculated across all stochastic simulations (10,000). To examine the influence of initial population size on the projected success rate, the model was run across a range of initial release sizes ranging from 1 to 20 bird pairs. We then selected the minimum release population size that led to an >80% chance of resulting in a stable or increasing population as the input for sensitivity analyses. We explored how variations in long-term and first-year adult survival rates and juvenile survival rates affect the probability of success in the 20-year period through several sensitivity analyses.

### 2.7. Web Application

In order to empower managers to consider the trade-offs in habitat area and quality associated with our integrated LHSM and PVA, we developed an interactive decision-making tool. We used the Shiny web application package in R to create a web-based tool for filtering potential release sites based on personalized criteria [44]. The known home range of adult 'ākohekohe is ~0.5 ha, with little evidence of overlap and strong evidence of strictly defended territories [13,14,20]. With this knowledge, we estimated how many 'ākohekohe adult individuals could be supported by each homogenous forest cluster by calculating the cluster area in hectares and multiplying those values by 2. We also incorporate the results of the PVA into the web application by representing the probability of viability for the maximum population supported by each cluster. These calculations allow for the efficient integration of our PVA results as well as the input of managers into CI planning.

## 3. Results

### 3.1. 'Ākohekohe on the Island of Hawai'i

The 'ākohekohe habitat suitability models with our curated list of vegetation structure and topography metrics performed well, with an average AUC score amongst 100 sub-models of 0.891. The standard deviation amongst these 100 iterations was 0.013, indicating a low level of uncertainty. Canopy height was the metric with the highest average percent contribution (40.8%) over replicate runs of the 'ākohekohe models (Table A4). Relative

height 90 (RH 90) had a high percentage contribution (30.9%) and the highest permutation importance (37.8%). RH 90 was also the variable with the highest regularized training gain when used in isolation, and decreased the gain the most when it was omitted (Figure A1). Response curves show that the effect of canopy height on the suitability prediction peaks at ~10 m (Figure A2a), whereas the dependence of predicted suitability on canopy density peaked around 75% (Figure A2b).

### 3.2. Candidate Site Evaluation

The Ka'ū candidate CI site had the greatest area that could be classified as forest based on the FAO definition (~28 km²), as well as the highest mean 'ākohekohe habitat suitability score within that forested area, with moderate mean uncertainty (0.77 ± 0.18) (Figure 3). Pu'u Maka'ala had the lowest mean 'ākohekohe habitat suitability score of all the candidate CI sites (0.21 ± 0.12). When incorporating the mean uncertainty (standard deviation amongst 100 sub-models) of the habitat suitability predictions at each candidate site, the mean suitability of Pu'u Maka'ala fell below that of Hakalau, Ka'ū, and Pu'u Wa'awa'a (Figure 3). Kīpuka had the second lowest average suitability amongst the sites (0.35 ± 0.15). Habitat suitability results at Pu'u Wa'awa'a had the highest average uncertainty (0.19).

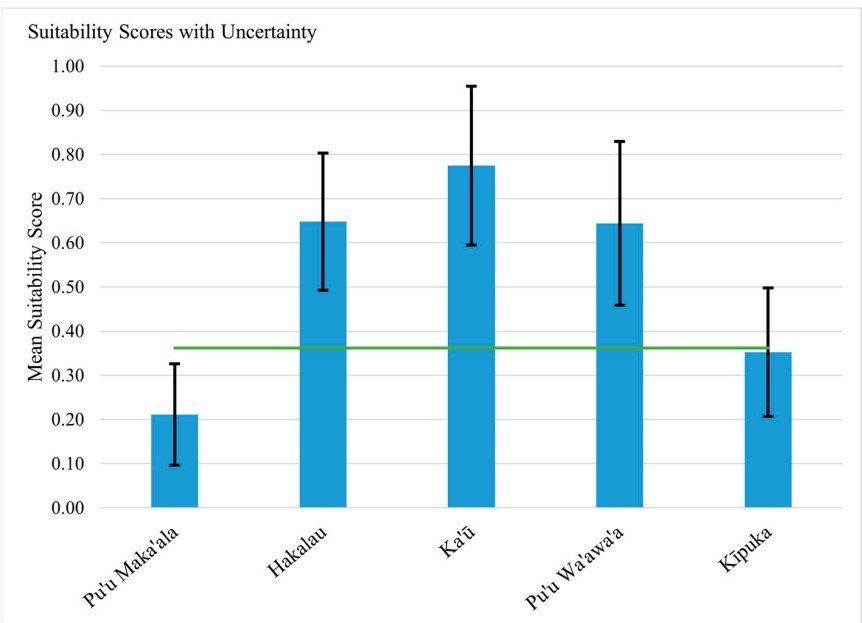

**Figure 3.** The average 'ākohekohe habitat suitability score by candidate CI site, represented by the blue bars. The black lines represent the range of average suitability when considering the standard deviation of the 100 sub-model habitat suitability predictions for each site. The green line represents the MTP threshold of 'ākohekohe suitability, as determined by Gallerani et al. (2023) [22].

We summarized the habitat suitability score and uncertainty of each 100 m pixel within the candidate CI sites through a set of maps (Figures 4 and A3). The HFAs on the suitability map display clusters of forests with homogenous suitability scores as discrete candidate CI release sites. The results of the GLCM homogeneity texture analysis without the application of a threshold are detailed in Appendix A (Figure A4). Upon visual inspection, areas that were the most suitable and homogenous were concentrated on the northern edge of Pu'u Wa'awa'a, the southeastern portion of Hakalau, and throughout Ka'ū (Figure 4). Hakalau had the most homogenous forested clusters (21) that also fell above the MTP threshold for 'ākohekohe habitat suitability (0.362) determined by Gallerani et al. (2023) [22] (Table 2). The combined area of these suitable clusters in Hakalau was ~3 km². Ka'ū had the second highest number of suitable clusters, with the average size of those clusters being around 10 times as large as the average cluster size in Hakalau, resulting in a combined suitable

area of ~19 km². All of Pu'u Maka'ala's 11 homogeneous forested clusters fell below the MTP threshold of 'ākohekohe suitability. Even with the exclusion of elevation from the model, the suitable habitat for 'ākohekohe was not found at elevations above ~1903 m, with the median elevation for suitable habitat clusters being ~1740 m.

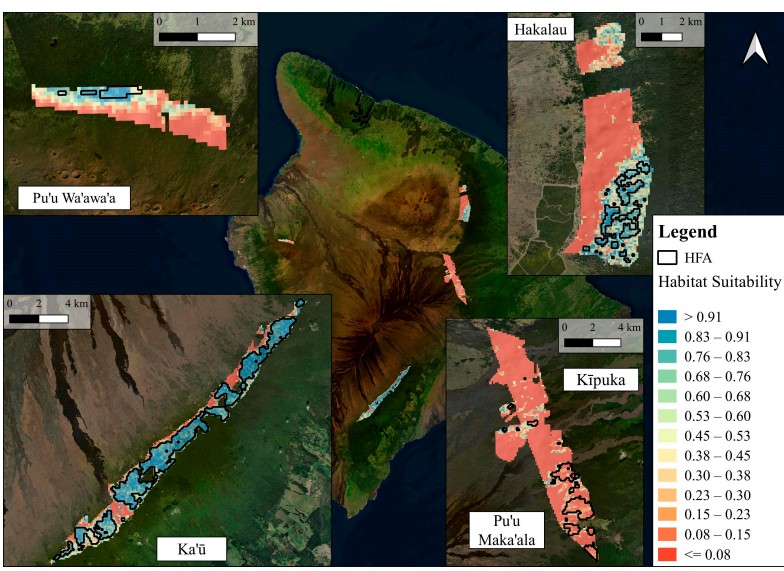

**Figure 4.** Average 'ākohekohe habitat suitability of each 100 m pixel across 100 sub-models, with red representing low suitability and blue representing high suitability. Forested areas where the suitability results are connected and homogenous are outlined by the black polygons at each site.

**Table 2.** Summary of homogenous forested areas clusters in each candidate CI site based on their suitability score as 'ākohekohe habitat. Clusters with an average suitability ≥ 0.362 were suitable; those below that score were not suitable. This table also contains the average size of suitable homogenous forested area clusters per candidate CI site.

| Candidate Sites | Suitable Clusters | Average Area (km²) | Total Area of Clusters (km²) | Not Suitable Clusters |
|---|---|---|---|---|
| Hakalau | 21 | 0.15 | 3.14 | 1 |
| Ka'ū | 13 | 1.48 | 19.22 | 2 |
| Pu'u Maka'ala | 0 | NA | 0 | 11 |
| Pu'u Wa'awa'a | 3 | 0.11 | 0.32 | 0 |
| Kīpuka | 2 | 0.04 | 0.09 | 5 |

### 3.3. Population Viability Analysis

Results from the initial deterministic individual run of our demographic model, as well as more detailed results from stochastic runs and sensitivity analyses, are presented in Appendix B. When exploring the effect of the starting population on the success rate, we found that after 1000 stochastic runs, at least nine pairs of birds are necessary for the release to have an >80% chance of resulting in a stable or increasing population. Additionally, at a release population size of eight pairs, there was a less than 1% chance of complete failure or extirpation (100% mortality) within the first 20 years after release. Our sensitivity analysis on long-term adult survival found that below 91% long-term adult survival after the initial year post release, the chance of CI success drops below 50%. When juvenile survival deviates below 0.20, the success rate drops below 50%. Success rates varied little based on adult survival for the first year post release (Appendix B). Beyond using the PVA to identify a minimum viable cluster size for CI, and the overall probability of success of CI for the species, we also used PVA to calculate the probability of population stability by population size of naturalized populations. This allowed us to estimate the population stability of individual clusters based on individual cluster carrying capacity (Appendix B).

### 3.4. Web Application

The HFAs, as discrete candidate CI release sites, are displayed in a web application with attributes of the mean habitat suitability, area in km$^2$, the number of adult 'ākohekohe individuals that could be supported by said area, and the probability of population stability given the PVA results. This application can be found at the following link: https://rconnect.usgs.gov/akohekohe_CI_tool/. Managers can scroll over the clusters in each of the five candidate CI sites to see these attributes. Additionally, they can select a suitability threshold from an interactive slider, along with the number of 'ākohekohe adults they wish to release that could lead to a viable population in the release site. The sites that meet these criteria will then be displayed on the map with different colors to represent average suitability. This allows managers to see how our results change by modifying specific criteria of potential release population success. If a manager wanted to explore clusters that could support 18 adult 'ākohekohe, then 26 clusters throughout all five candidate sites have a large enough area to support such a population. That number would continue to decrease depending on the threshold for 'ākohekohe suitability.

## 4. Discussion

### 4.1. Lidar Sensor Correction

One significant factor likely contributing to any model uncertainty in this study is the diversity of sensors used to collect lidar data across the candidate CI sites. We tried to reduce this uncertainty through our sensor correction procedure. Previous research has found that when considering appropriate factors, the combination of data from various lidar projects does not hinder the accurate estimation of canopy structure [29]. We compared sensor specifications that would most likely lead to differences in point clouds collected over similar areas. Beam divergence indicates how wide or narrow the laser pulse spreads as it travels from the sensor to the Earth's surface. A wider (narrower) beam divergence can result in a larger (smaller) footprint on the ground. This works in concert with the altitude at which the sensor is flown, which controls the resolution of the data, and therefore, the coverage on the ground per laser pulse [45]. Depending on these two parameters, sensors can collect either finer or coarser point clouds over the same area [46,47]. Similarly, the pulse repetition frequency (PRF) affects the density of point clouds. PRF is the rate at which a sensor emits laser pulses, typically measured in Hertz (Hz) [40]. This parameter allows sensors to capture more (or fewer) data points per unit of time. Finally, the wavelength of light used by the sensor will determine how well the laser light can penetrate the canopy and capture lower vegetation and ground points [48]. Differing wavelengths could therefore greatly affect the comparison of forest structure metrics such as RH 25 between different sensor point clouds in the same forested area. Considering these relevant parameters, we were able to create relationships to standardize the data, and thereby reduce uncertainty in our model results. Furthermore, the mixed ground cover types of the overlap areas between sensors ensured that our regression equations were applicable in both dense canopies and more open landscapes.

### 4.2. Habitat Preferences

The relative importance of canopy height, RH 90, and canopy density to the model performance can shed light on habitat preferences of 'ākohekohe. It is important to note that despite our efforts to eliminate highly correlated variables, RH 90, canopy density and canopy height do experience high correlations. Therefore, it is difficult to parse out the individual metric importance. However, the effect of RH 90 on the ability of the model to predict suitable 'ākohekohe habitat suggests the importance of upper canopy forest structure in habitat selection for this species. RH 90 more specifically encapsulates the top of canopy structure than the canopy height model, and it can be representative of forest biomass [49,50]. It is also highly correlated to RH 75 which was used in previous research to represent the majority of overstory vegetation [23]. In addition to evaluating the importance of individual metrics, we can assess how the range of each metric's values affects the model

predictions using response curves. The ideal forest structure for 'ākohekohe suitability based on our model at the 100 m scale was ~10 m average tree height, a canopy density of ~75%, an understory height (RH 25) of ~2 m, and a tall upper canopy with RH 90 peaking at ~15 m (Figure A2a–d). This is corroborated by a previous observational study of 'ākohekohe foraging, which found that they foraged in the canopy during 64% of all observations at a mean height of 9.5 m $\pm$ 0.90. Foraging from subcanopy trees and shrubs was seasonal, and mainly occurred when 'ōhi'a bloom declined [13]. These preferences highlighted by our model results show that the management of canopy tree species is essential for the success of translocated 'ākohekohe. Evaluation and monitoring of rapid 'ōhi'a death (ROD) at any potential release sites would help ensure the continued quality of the candidate site [51]. The timing of 'ākohekohe breeding has also been shown to be positively correlated with a high abundance of 'ōhi'a lehua blossoms. Therefore, quantitative analysis of 'ōhi'a phenology in these candidate CI sites could be incorporated as an assessment of CI potential.

It Is important to note that the habitat preferences elucidated by this model reflect 'ākohekohe behavior in its current restricted range on east Maui. Historically, 'ākohekohe were found in a wider range over Maui, and even on the island of Moloka'i [11]. We therefore cannot definitively rule out some shift in vegetation structure preferences over the past century, given their range limitations due to changing disease and climate landscapes. However, 'ākohekohe appear specialized to particular forest types and vertical stratification within native wet forests; such traits are not likely to have shifted greatly in a matter of decades. Assuming historical stands occupied by 'ākohekohe shared broad structural similarities to current forest, our models have likely captured the key structural elements of suitable habitat despite the more recent input data. Using the forest structure decoupled our results from confounding temporal factors, allowing us to elucidate effective suitable habitats in the absence of disease.

### 4.3. Candidate Conservation Introduction Site Comparison

Considering suitability scores, homogeneity, and uncertainty, Ka'ū and Hakalau were the top candidate CI sites amongst the five sites considered. Pu'u Wa'awa'a followed closely behind Hakalau in average habitat suitability score; however, the site did have a higher average uncertainty amongst results of the 100 sub-models. Additionally, it was quite a small site with only 0.32 km$^2$ of total suitable and homogenous 'ākohekohe habitat. Although Hakalau had the highest number of discrete suitable 'ākohekohe habitat clusters, Ka'ū had the largest area of homogenous suitable habitat. During the 2022 SWOT analysis, experts gave the best habitat indicator scores for 'ākohekohe to Hakalau when considering species-specific indicators [17]. Hakalau also ranked higher in quality than Ka'ū in several crucial categories, such as the current limited presence of threats such as avian malaria, pigs, and mosquito breeding habitat. Quantitative analyses to defend these expert rankings are essential to determine which candidate CI site is truly more suitable. However, strictly based on forest structure, Ka'ū is the most species-specific suitable candidate CI site for 'ākohekohe. Other habitat qualities conducive to 'ākohekohe success, such as the presence of specific understory plant species, which provide nectar or major arthropod prey, may not be fully captured by our model. Weighing additional habitat characteristics and potential threats would allow for more confidence when choosing a CI site. Specifically, more work would be beneficial to determine the presence and potential future presence of avian malaria at these sites.

Based on our LHSM results, Pu'u Maka'ala had the lowest average suitability score of all candidate CI sites (Figure 3). In addition, Pu'u Maka'ala had no homogenous areas above the MTP threshold for 'ākohekohe (Table 2). We would expect that forested area in a candidate CI site would follow a close relationship to the total area of suitable 'ākohekohe habitat. This was certainly true for Ka'ū, the site with the largest forested area (28.41 km$^2$) and most homogenous suitable 'ākohekohe habitat (19.22 km$^2$). However, Pu'u Maka'ala had the third largest forested area (7.38 km$^2$) of the candidate sites and the least amount of suitable habitat (Table 2). The fact that there were non-suitable homogenous forested

clusters indicates that the determination of species-specific suitable forest structures is more complex than simple metric thresholds. In fact, the complex relationships between relative height metrics, canopy density, canopy height, and even topography seem to drive 'ākohekohe habitat suitability across our examined sites. Our results emphasize the importance of the quantitative analysis of forest structure and composition to determine suitable habitats for avian species CI both in Hawai'i and globally. Furthermore, caution is warranted when considering Pu'u Maka'ala for 'ākohekohe release, despite its high scores in the 2022 SWOT analysis, as the forest structure appears unsuitable based on our results.

*4.4. Release Site Selection*

The substantial benefit of our 100 m analysis is the determination of suitable release sites within the larger candidate CI areas. With clusters that met specific homogeneity and suitability criteria, we were able to identify specific areas fit for the release of 'ākohekohe individuals. One caveat of our cluster-based analyses is that every cluster is considered independent regardless of their proximity to other clusters. Some of these smaller clusters in the Hakalau site, for example, may be functionally connected for birds moving across landscapes. Our results are still useful in identifying areas in which released populations would minimize movement across unsuitable habitat. However, additional inquiry into the degree of risk incurred by traveling short distances over less suitable habitat could be considered in the future when selecting release sites.

A key element in determining the appropriateness of a release site is if the area is large enough to support a population that can persist many years into the future. After several simulations, our demographic model results indicated that nine pairs is an initial release population of 'ākohekohe that is robust against stochasticity and will likely result in a stable or growing translocated population. Although the PVA provided a useful initial estimate of release size and 'ākohekohe CI viability, these results warrant careful interpretation given the underlying data limitations and biological assumptions. Several key demographic parameters, especially juvenile survival, are based on estimates from other Hawaiian honeycreeper species due to the lack of empirical data for 'ākohekohe specifically. Furthermore, the model does not account for potential density dependence effects on vital rates as the population grows. The sensitivity analyses revealed adult survival and first-year effects as critical knowledge gaps. The CI success appeared highly contingent on maintaining high adult survival rates typical of wild 'ākohekohe populations on Maui. If CI substantially reduces long-term adult survival, the population growth may be insufficient to sustain the population, as the models currently predict. However, when a much lower bar for success was applied to these results, the likelihood of complete extirpation of the population within 20 years was much less sensitive to variability in adult and juvenile survival. A further discussion of caveats and uncertainties is presented in Appendix B.

Our web application allows for the interaction of managers with the results of our integrated LHSMs and PVAs and serves as a model for future applications of our methodology. The application also makes it possible to see how refined release size estimates would reshape our suitability results. One limitation of our web application is that we only consider adult home ranges. There is strong evidence that 'ākohekohe juveniles travel long distances during the post-fledging and pre-breeding periods, and occupy areas closer to 25.02 ha in size [14]. This may indicate that non-territorial juveniles require a much larger area than adults. However, because the juveniles are not territorial during this period, they exhibit a great deal of overlap in the habitats within which individuals occupy [14]. A high degree of overlap makes the estimation of the total required release area more challenging. However, the consideration of the surrounding habitat suitability when selecting release sites would help to ensure the ability of juveniles to travel and establish new territories. This also further highlights the potential importance of contiguous habitats and/or a matrix of habitat patches that facilitates adequate juvenile dispersal.

## 5. Conclusions

This study demonstrates the value of high-resolution lidar data for assessing habitat compatibility to support endangered species CI. By standardizing vegetation structure metrics across differing lidar sensors in the native and potential introduction ranges, habitat suitability models were successfully transferred between islands with high AUC scores. Our methods for sensor correction provide a useful guide to working with lidar data on larger landscapes given the patchwork of sensors and parameters used for collection in Hawai'i and elsewhere. The LHSM revealed distinct habitat preferences of 'ākohekohe that transcend beyond simple forest structure metric thresholds. The dependence of the 'ākohekohe habitat suitability scores on average canopy height and canopy density peaked at ~10 m and 75%, respectively. Compared with a previous study [22], we took our research further through our consideration of the spatial heterogeneity of suitable habitat. We thus identified patches of consistent suitable habitat using a texture analysis of model results across the landscape rather than relying on site-wide averages. Ka'ū proved to be the candidate site with the most suitable habitat, while Pu'u Maka'ala was found to be the least suitable. However, the consideration of several other abiotic, biotic, logistical, and management factors is essential when selecting a CI site. Initial PVA results show that a minimum of nine pairs is necessary for establishing a successful translocated population of 'ākohekohe. The web application provides an interactive decision support tool for the consideration of PVA caveats when selecting viable founder population release sites. The methodology developed here establishes a framework to quantitatively evaluate habitat suitability and connectivity for CI of any species dependent on forest structure and topography globally.

**Author Contributions:** Conceptualization, E.M.G. and L.B.F.; methodology, E.M.G., L.B.F., and C.C.W.; data analysis, E.M.G. and L.B.F.; validation, C.C.W. and E.H.P.; writing—original draft preparation, E.M.G. and L.B.F.; writing—review and editing, C.C.W. and E.H.P.; visualization, E.M.G. and L.B.F. All authors have read and agreed to the published version of the manuscript.

**Funding:** This research received USGS Ecosystems Mission Area funds for its completion.

**Data Availability Statement:** All lidar-derived environmental layers for Maui and the Island of Hawai'i can be found at https://doi.org/10.5066/P1CEQA9X [52]. Global positioning system (GPS) locations of endangered bird species are not available publicly given their sensitivity. These data can be requested by reaching out to the corresponding author.

**Acknowledgments:** We would like to thank members of the U.S. Fish and Wildlife Service SWOT analysis team, and specifically Paul Banko, who contributed feedback to our initial results and writing. We are grateful to the Maui Forest Bird Recovery Project crew who collected all field data of 'ākohekohe locations. Any use of trade, firm, or product names is for descriptive purposes only, and does not imply endorsement by the U.S. Government.

**Conflicts of Interest:** The authors declare no conflicts of interest.

## Appendix A

**Table A1.** Comparison of sensor specifications from different aerial lidar data. Abbreviations used: AGL, above ground level; GAO, global airborne observatory; NSF, National Science Foundation; USGS, U.S. Geological Survey; FWS, U.S. Fish and Wildlife Survey.

| Source | Sensor Name | Beam Divergence (mrad) | Altitude (m AGL) | PRF (kHz) | Wavelength (nm) |
| --- | --- | --- | --- | --- | --- |
| GAO | Optech HA-500 | 0.25 | 2000 | 100–500 | 1064 |
| NSF | Optech Gemini | 0.25–0.8 | 1000 | 100 | 1064 |
| USGS | Leica SPL | 0.08 | 3200–4420 | 60 | 532 |
| FWS | Riegel LMS-Q680i | ≤0.5 | 500 | 80–400 | Near-infrared (700–1400) |

**Table A2.** Regression coefficients and adjusted R-squared values of the linear relationships between overlapping lidar data collected from different sensors on the Island of Hawai'i. These coefficients were then used to sensor-correct the different lidar data to better match data collected by the Optech Gemini sensor implemented by NSF. All *p*-values of the regression models were $<2 \times 10^{-16}$.

| Metrics | Hakalau (USGS~FWS) | | | Pu'u Maka'ala (USGS~NSF) | | |
|---|---|---|---|---|---|---|
| | Intercept | Slope | $R^2$ | Intercept | Slope | $R^2$ |
| Canopy Height | −0.22 | 0.82 | 0.94 | 0.44 | 0.87 | 0.94 |
| Relative Height$_{25}$ | 0.21 | 1.16 | 0.54 | −0.02 | 0.33 | 0.81 |
| Relative Height$_{50}$ | 1.15 | 0.65 | 0.82 | 0.16 | 0.81 | 0.92 |
| Relative Height$_{75}$ | 0.23 | 0.74 | 0.92 | 0.55 | 1.00 | 0.93 |
| Relative Height$_{90}$ | −1.21 | 0.80 | 0.91 | 0.69 | 1.09 | 0.93 |
| Canopy Density | −3.20 | 0.89 | 0.85 | 6.51 | 1.02 | 0.89 |

**Table A3.** Spearman rank correlation analysis of nine lidar-derived vegetation structure and topographic metrics in the candidate conservation introductions (CI) areas on the Island of Hawai'i. Abbreviations used: CHM, canopy height model; DNS, canopy density; DTM, digital terrain model (elevation); RH, relative height (25, 50, 75, and 90 represent the percentages of returns that fall below the relative height); TWI, topographic wetness index.

| | CHM | DNS | DTM | RH25 | RH50 | RH75 | RH90 | SLOPE |
|---|---|---|---|---|---|---|---|---|
| CHM | 1.00 | | | | | | | |
| DNS | 0.89 | 1.00 | | | | | | |
| DTM | −0.04 | −0.15 | 1.00 | | | | | |
| RH$_{25}$ | 0.82 | 0.88 | −0.16 | 1.00 | | | | |
| RH$_{50}$ | 0.98 | 0.88 | −0.02 | 0.82 | 1.00 | | | |
| RH$_{75}$ | 0.98 | 0.84 | −0.01 | 0.75 | 0.97 | 1.00 | | |
| RH$_{90}$ | 0.97 | 0.83 | −0.02 | 0.73 | 0.94 | 0.99 | 1.00 | |
| SLOPE | 0.40 | 0.31 | 0.23 | 0.19 | 0.41 | 0.46 | 0.47 | 1.00 |
| TWI | −0.14 | −0.09 | −0.27 | 0.05 | −0.17 | −0.21 | −0.21 | −0.67 |

**Table A4.** Variable percent contribution and permutation importance of the 'ākohekohe (*Palmeria dolei*) habitat suitability models.

| Variable | Percent Contribution | Permutation Importance |
|---|---|---|
| Canopy Height | 40.8 | 18.7 |
| RH$_{90}$ | 30.9 | 37.8 |
| Canopy Density | 20.9 | 18.9 |
| TWI | 4.5 | 3.9 |
| RH$_{25}$ | 2.9 | 20.7 |

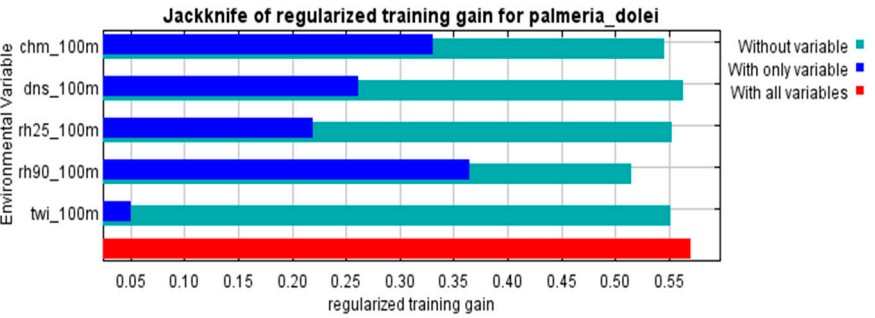

**Figure A1.** Jackknife or regularized training gain plot for 'ākohekohe (*Palmeria dolei*) habitat suitability. This graph displays how the model performance changes when a model is built using just one environmental variable or when one environmental variable is excluded. All variables have the suffix "_100 m" to represent the scale of the environmental variables.

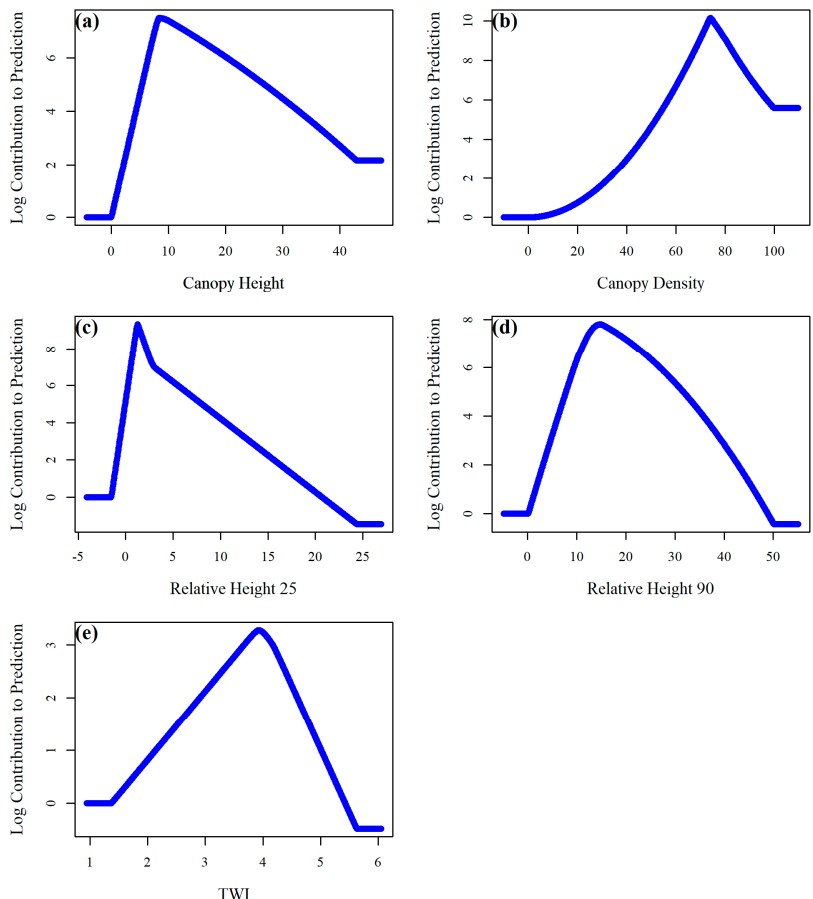

**Figure A2.** Response curves, each representing a different model created using only the corresponding lidar-derived metric at the 100 m scale. These charts illustrate how the predicted suitability of 'ākohekohe (*Palmeria dolei*) is influenced by the chosen variables and the correlations they share with other variables. The metrics are as follows: canopy height (**a**), canopy density (**b**), relative height 25 (**c**), relative height 90 (**d**), topographic wetness index (**e**).

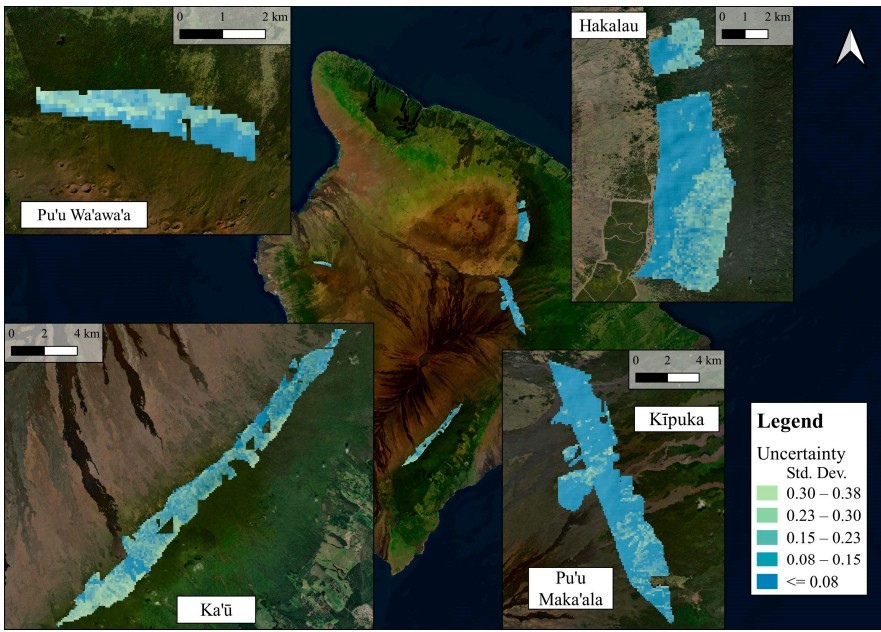

**Figure A3.** Pixel-wise standard deviation of the 100 models of 'ākohekohe (*Palmeria dolei*) habitat suitability applied to the lidar metric layers for the Island of Hawai'i.

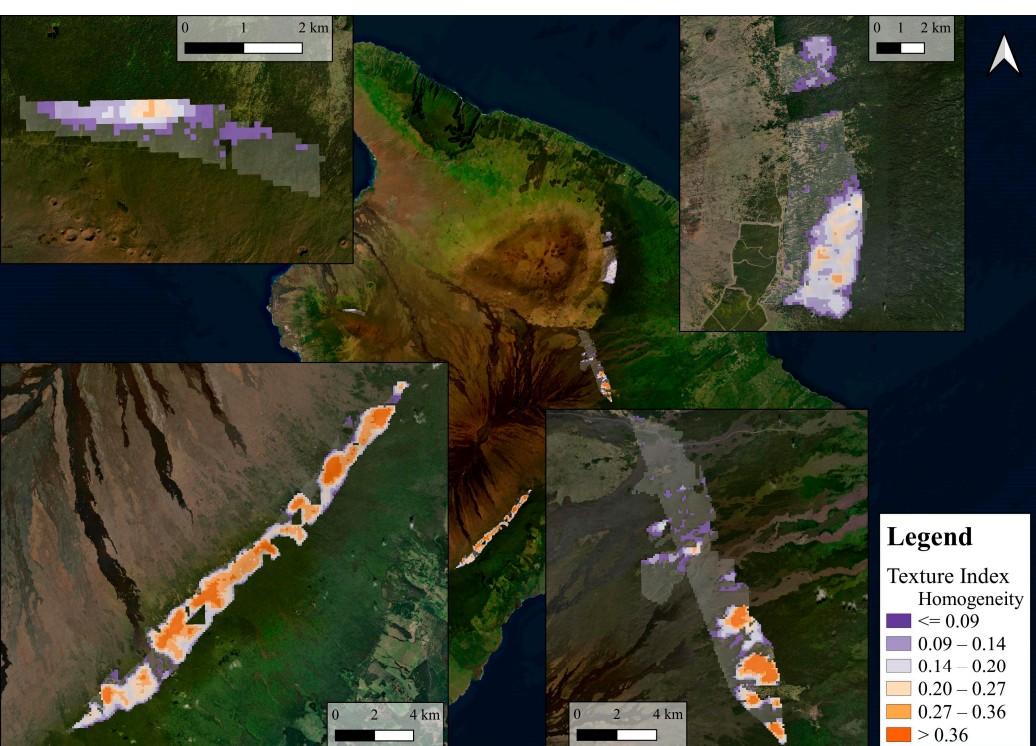

**Figure A4.** Results of the 3 × 3 gray-level co-occurrence matrix texture statistic as applied to 'ākohekohe (*Palmeria dolei*) habitat suitability scores at the 100 m scale in areas that meet the Food and Agriculture Organization of the United Nations (FAO) forest definitions. Purple areas represent more heterogeneous habitat scores; orange areas represent more homogenous habitat scores.

## Appendix B. Population Viability Analysis for 'Ākohekohe (*Palmeria dolei*) Conservation Introductions on the Island of Hawai'i

To estimate the minimum number of birds required for a conservation introduction (CI) release, we performed population viability analysis (PVA) using stochastic simulations in R. The model was developed considering the available demographic parameters for the species based on previous research. Simulations focused on determining the number of bird pairs needed per release; however, as often conducted with monogamous animal species [53,54], the calculations were based on the number of female birds, assuming an equal sex ratio in the releases. Several demographic parameters were integrated into the model, detailed subsequently. **Pairs Renesting within Season ($p$):** The proportion of pairs that would attempt to renest within a single breeding season was set to 0.42 [55]. **Nest Success ($s$):** The probability of a nest being successful was allowed to vary uniformly between 0.59 and 0.83, which are low and high yearly estimates from field observations [55]. **Chicks per Nest ($c$):** The average number of chicks produced per successful nest was set to 1.1 [55]. **Juvenile Survival Rate ($s_j$):** The juvenile survival rate is a key demographic parameter influencing population viability. However, limited data are available on juvenile survival specifically for the 'ākohekohe. To derive a reasonable estimate, we performed weighted mean calculations using available juvenile survival data from other Hawaiian honeycreeper species [56]. This resulted in an initial juvenile survival mean of 0.2526 and standard error of 0.0507. Given that the 'ākohekohe's closest relatives, the i'iwi (*Vestiaria coccinea*) and 'apapane (*Himatione sanguinea*), have markedly lower juvenile survivorship, we also initially considered a lower bound estimate calculated only for these two species (mean, 0.1107; standard error (SE), 0.0604). However, unpublished observational data from Wang et al. (2020) [14] showed a much high juvenile survivorship for 'ākohekohe. Therefore, we opted to utilize the weighted mean juvenile survival rate calculated across all honeycreeper species (mean, 0.2526; SE, 0.0507) in our primary analysis. We conducted sensitivity analyses based on lower mean juvenile survival rates to further explore the

influence of this important parameter. The probability of a juvenile surviving was modeled to stochastically range normally around the mean of 0.25 with an SE of 0.05. To avoid probability values below 0 or above 1, we used truncated normal distribution to sample values from year to year using the R package truncnorm. **Adult Survival Rate ($s_a$):** The survival probability of adults was allowed to stochastically range normally around a mean of 0.95 with an SE of 0.1 [55]. To avoid probability values below 0 or above 1, we used a truncated normal distribution to sample values from year to year using the R package truncnorm. **The initial number of adult females ($N_{f,0}$):** This value was varied between 1 and 30 across different simulations to assess the impact of release size on the future trajectory of the translocated population. **Impact of conservation introduction on population demography:** We assumed that the first year after release, adult survival probabilities would be lower due to the stress of captivity, transportation, and adaptation to a new environment. Following consultation with experts in prior bird CIs, this effect was simply simulated as a 10% decrease in mean annual survival of adults. We also assumed no successful nesting in the first year following release; thus, there was no need to define the impact of CI on juvenile survival.

The population was projected over a 20-year period using a discrete-time model, encapsulating the demographic parameters and their inherent stochasticity. For each year ($t$), the following computations were executed:

$$Nests\ Produced_t = Adults_t + (Adults_t \times p)$$

$$Nest\ Sucess_t = Uniform(s_{lower}, s_{upper})$$

$$Nests\ Successful_t = Nests\ Prodcued_t \times Nest\ Success_t$$

$$Juveniles\ Produced_t = Nests\ Successful_t \times c$$

$$Juvenile\ Survival\ Rate_t = Truncated\ normal(s_j)$$

$$Juveniles\ Surviving_t = Juveniles\ Produced_t \times Juvenile\ Survival\ Rate_t$$

$$Juvenile\ Females\ Surviving_t = Juveniles\ Surviving_t / 2$$

$$Adult\ Survival\ Rate_t = Truncated\ normal(s_a)$$

$$Adult\ Survival\ Rate_t = Truncated\ normal(s_a)$$

$$N_t = Juvenile\ Females\ Surviving_t + Surviving\ Adult\ Females_t$$

where $Uniform(a, b)$ generates a random number following a uniform distribution between $a$ and $b$, and $Truncated\ normal(a)$ generates a random number following a truncated normal distribution around $a$ using the available standard error estimate for the parameter. The number of adults in the subsequent year was determined by the value of $N_t$, forming the basis for the calculations in the subsequent year.

*Appendix B.1. Single Run and Stochastic Simulations*

A single run of the model provided a deterministic trajectory of the population, offering insight into the possible future state of the population under specified conditions. For these individual runs, the surviving numbers of individuals were rounded to ensure that demographic stochasticity would be a factor in the projections. To account for the inherent stochasticity and uncertainty in the demographic parameters, 1000 stochastic simulations were executed, each generating a unique population trajectory over the 20-year period. The stochasticity was incorporated by allowing the adult and juvenile survival rates and nest success to vary randomly within specified bounds. The model was based on modeling the females only, and an additional source of demographic stochasticity was added by using a binomial distribution to determine the number of females born out of the total number of juveniles born in each year of each individual simulation. This was thought to be an important source of demographic stochasticity [57] given the generally small

number of bird pairs considered for a potential CI release. Furthermore, to avoid potential bias when modeling small populations, we incorporated stochastic rounding whenever simulating discrete values such as number of nests or surviving adults and juveniles. Deterministic nearest rounding could, for instance, result in a situation where populations never dropped below 1 individual, as with just 1 female remaining, a 95% $\pm$ 0.1 SE adult survival rate would always round up to 1 surviving female. Rather than deterministic rounding, we implemented stochastic rounding so that a value such as 1.6 successful nests would have a 60% chance of rounding down to 1 and 40% chance of rounding up to 2. This introduced additional demographic stochasticity that is characteristic of small populations. Specifically, stochastic rounding was applied when calculating the rounded number of nests produced, successful nests, surviving juveniles, surviving adults, and total remaining population each year. This helps ensure that extinction is possible in the model even when multiplying very small populations by high survival rates.

A successful run was defined as a simulation where the population was stable or increasing over the 20-year simulation (i.e., the population size at year 20 was equal or greater than the initial population size). The proportion of successful runs was calculated across all stochastic simulations (10,000), providing an estimate of the probability of the population remaining stable or increasing over the 20-year period under the defined conditions.

*Appendix B.2. Sensitivity Analysis*

To examine the influence of initial population size on the projected success rate, the model was run across a range of initial female population sizes from 1 to 20. For each starting population size, the model was run 1000 times and the success rate was computed, facilitating the exploration of the relationship between initial population size and future population viability.

*Appendix B.3. R Code and Results*

This section presents all relevant R scripts necessary to replicate the PVA embedded within the text, to allow for easier replicability of the analysis and associated findings.

Appendix B.3.1. Defining Constants

Initially, we defined constants for our PVA model. These constants represent estimates and assumptions about the 'ākohekohe bird species and were used in our single and multiple run simulations [55].

```
pairs_renest_within_season <- 0.42 #(simon et al 2001)
chicks_per_nest <- 1.1 #(simon et al 2001)
juvenile_survival_mean <- 0.2526 #(woodworth and pratt 2009) #weighted mean for all creepers
juvenile_survival_se <- 0.0507 #(woodworth and pratt 2009)  #weighted mean for all creepers
adult_survival_mean <- 0.95 #(simon et al 2001)
adult_survival_se <- 0.1 #(simon et al 2001)
nest_success_upper <- 0.83 #monitored year with highest rate; with 0.68 being both the median ye
ar and the overall mean (simon et al 2001)
nest_success_lower <- 0.59 ##monitored year with lowest rate (simon et al 2001)

firstyr_juvenile_survival_effect <- -0.1 #this does not impact the model now but could if we wer
e to make the translocation effect longer than 1 year
firstyr_adult_survival_effect <- -0.1
firstyr_nest_success_upper <- 0
firstyr_nest_success_lower <- 0

starting_N_females <- 9
num_years <- 20
num_simulations=10000
```

Appendix B.3.2. Creating Model for Individual Release Population Projection

First, we created a function, run_population_projections, which generates a single run population projection based on the demographic variables defined above.

```
set.seed(321)

#first create a function to stochastically round individuals
#Stochastic rounding means that a number is rounded up or down based not simply on its fractiona
l part, but on a probabilistic rule. For example, the number 4.2 would have an 80% chance of bei
ng rounded down to 4 and a 20% chance of being rounded up to 5.
#this is important because if we use simple rounding, when we split 3 juveniles using a 0.5 mal
e/female ratio, rounding of 1.5 will always lead to 2.

stochastic_round <- function(x) {
  # Get the fractional part
  frac_part <- x - floor(x)
  # Stochastically determine whether to round up or down
  if (runif(1) < frac_part) {
    return(ceiling(x))
  } else {
    return(floor(x))
  }
}
# Test the function
#stochastic_round(1.5)

#define function for creating a single population projection
run_population_projection <- function(pairs_renest_within_season, chicks_per_nest,
                                      juvenile_survival_mean, juvenile_survival_se,
                                      adult_survival_mean, adult_survival_se,
                                      nest_success_upper, nest_success_lower,
                                      firstyr_juvenile_survival_effect,
                                      firstyr_adult_survival_effect,
                                      firstyr_nest_success_upper, firstyr_nest_success_lower,
                                      starting_N_females, num_years = 20, binomial_sexratio=T) {

  # Initialize a data frame to store results
  results <- data.frame(matrix(nrow = num_years, ncol = 12))
  colnames(results) <- c("Year", "Adults", "NestsProduced", "NestSuccess", "NestsSuccessful", "J
uvenilesProduced",
                         "JuvenileSurvivalRate", "JuvenilesSurviving", "JuvenileFemalesSurvivin
g",
                         "adult_survival_rate", "surviving_adult_females", "N")

  # Set initial state for year 1
  results$Year[1] <- 1
  results$Adults[1] <- starting_N_females

  # Model loop for num_years years
  for (year in 1:num_years) {
    if (year > 1) {
      results$Adults[year] <- results$N[year - 1]

      yr_juvenile_survival_mean=juvenile_survival_mean
      yr_nest_success_lower=nest_success_lower
      yr_nest_success_upper=nest_success_upper
```

```
      yr_adult_survival_mean=adult_survival_mean
    }else{
      yr_juvenile_survival_mean=juvenile_survival_mean+firstyr_juvenile_survival_effect
      yr_nest_success_lower=firstyr_nest_success_lower
      yr_nest_success_upper=firstyr_nest_success_upper
      yr_adult_survival_mean=adult_survival_mean+firstyr_adult_survival_effect
    }
    yr_juvenile_survival_se=juvenile_survival_se
    yr_adult_survival_se=adult_survival_se

    results$NestsProduced[year] <- stochastic_round(results$Adults[year] + (results$Adults[year]
* pairs_renest_within_season))
    results$NestSuccess[year] <- runif(1, yr_nest_success_lower, yr_nest_success_upper)
    results$NestsSuccessful[year] <- stochastic_round(results$NestsProduced[year] * results$Nest
Success[year])
    results$JuvenilesProduced[year] <- stochastic_round(results$NestsSuccessful[year] * chicks_p
er_nest)
    results$JuvenileSurvivalRate[year] <- rtruncnorm(1, a=0, b=1, mean=yr_juvenile_survival_mea
n, sd=yr_juvenile_survival_se) #we are using a truncated normal distribution since we do not kno
w the n and sd for the SE estimate provided
    results$JuvenilesSurviving[year] <- stochastic_round(results$JuvenilesProduced[year] * resul
ts$JuvenileSurvivalRate[year])
    if (binomial_sexratio){
      results$JuvenileFemalesSurviving[year] <- rbinom(1, results$JuvenilesSurviving[year], 0.5)
    }else{
      results$JuvenileFemalesSurviving[year] <- stochastic_round(results$JuvenilesSurviving[yea
r] / 2)
    }
    results$adult_survival_rate[year] <- rtruncnorm(1, a=0, b=1, mean=yr_adult_survival_mean, sd
=yr_adult_survival_se) #we are using a truncated normal distribution since we do not know the n
and sd for the SE estimate provided
    results$surviving_adult_females[year] <- stochastic_round(results$Adults[year] * results$adu
lt_survival_rate[year])
    results$N[year] <- stochastic_round(results$JuvenileFemalesSurviving[year] + results$survivi
ng_adult_females[year])
    results$Year[year] <- year
  }
  return(results)
}
```

### Appendix B.3.3. Running a Deterministic Individual Release Population Projection

We first applied the function above and visualized a single run of the population projection using the deterministic rates of adult survival (0.95) [55], juvenile survival (0.25) [56], and nest success (0.68) [55], based on rates extracted from the literature.

```
set.seed(321)
projection <- run_population_projection(
  pairs_renest_within_season, chicks_per_nest,
  juvenile_survival_mean, juvenile_survival_se=0,
  adult_survival_mean, adult_survival_se=0,
  nest_success_upper=0.68, nest_success_lower=0.68,
  firstyr_juvenile_survival_effect,
  firstyr_adult_survival_effect,
  firstyr_nest_success_upper, firstyr_nest_success_lower,
  starting_N_females, binomial_sexratio = T
)
plot(projection$Year, projection$N, xlab="Year", ylab="N females")
```

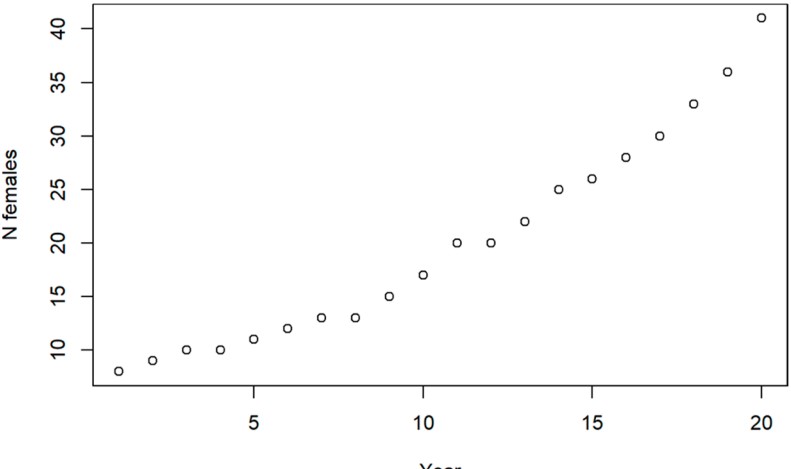

Please note in the graph above the irreducible stochasticity due to the binary nature of the sex ratios of individual birds born into the population each year, and due to the stochastic rounding described above.

Appendix B.3.4. Stochastic Release Population Projection

The next step was to create a function, simulate_population, which performs multiple runs (stochastic projection) and calculates the proportion of runs that result in a final population greater than the initial population.

```
simulate_population <- function(
    pairs_renest_within_season, chicks_per_nest,
    juvenile_survival_mean, juvenile_survival_se,
    adult_survival_mean, adult_survival_se,
    nest_success_upper, nest_success_lower,
    firstyr_juvenile_survival_effect,
    firstyr_adult_survival_effect,
    firstyr_nest_success_upper, firstyr_nest_success_lower,
    starting_N_females, num_years = 20, num_simulations = 10000, extirpation_risk=F) {

  successful_runs <- 0
  #sim=1
  for (sim in 1:num_simulations) {
    results <- run_population_projection(
      pairs_renest_within_season = pairs_renest_within_season, chicks_per_nest = chicks_per_nes
t,
      juvenile_survival_mean=juvenile_survival_mean, juvenile_survival_se=juvenile_survival_se,
      adult_survival_mean=adult_survival_mean, adult_survival_se=adult_survival_se,
      nest_success_upper = nest_success_upper, nest_success_lower = nest_success_lower,
      firstyr_juvenile_survival_effect=firstyr_juvenile_survival_effect,
      firstyr_adult_survival_effect=firstyr_adult_survival_effect,
      firstyr_nest_success_upper=firstyr_nest_success_upper, firstyr_nest_success_lower=firstyr_
nest_success_lower,
      starting_N_females = starting_N_females)
    # Check if the final population is larger than the starting population
    if (extirpation_risk){
      if (results$N[num_years] > 0) { #results$N[num_years] >= starting_N_females
        successful_runs <- successful_runs + 1
      }
    }else{
      if (results$N[num_years] >= starting_N_females) { #results$N[num_years] >= starting_N_fema
les
        successful_runs <- successful_runs + 1
      }
```

### Appendix B.3.5. Explore the Success Rate by Starting Population

Using the stochastic projection function defined above, we then explored how different initial population sizes affect the success rate of ʻākohekohe releases. This is the key result we used to estimate the minimum release size for the rest of the study.

```
set.seed(321) starting_N_females_range
<- 1:30
success_rates <- sapply(starting_N_females_range, function(starting_N_females) { simulate_population(
    pairs_renest_within_season, chicks_per_nest,
    juvenile_survival_mean, juvenile_survival_se,
    adult_survival_mean, adult_survival_se,
    nest_success_upper, nest_success_lower,
    firstyr_juvenile_survival_effect,
    firstyr_adult_survival_effect, firstyr_nest_success_upper,
    firstyr_nest_success_lower, starting_N_females, num_years,
    num_simulations
  )
})

plot(starting_N_females_range, success_rates, type = "b", pch = 19,
    xlab = "Starting Number of Females", ylab = "Success Rate", main
    = "Success Rate by Starting Population", ylim=c(0,1))
```

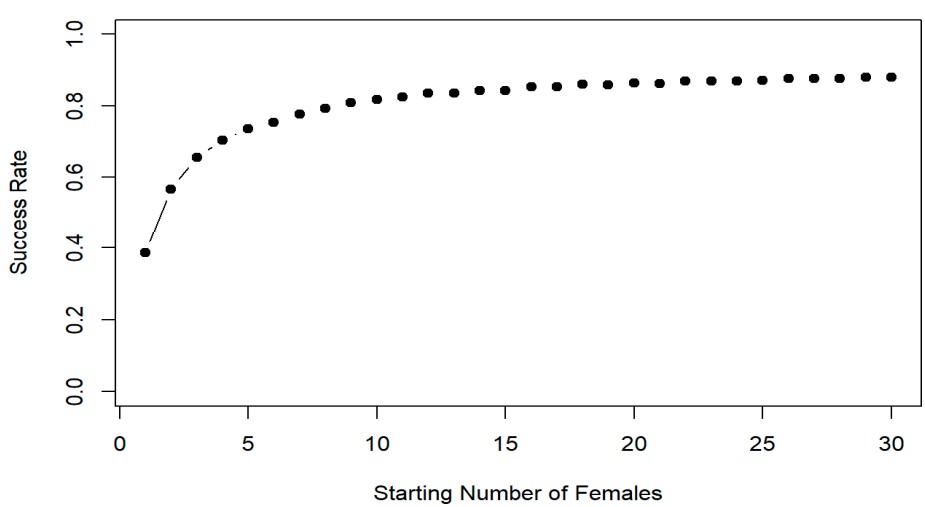

```
output_prob_success_by_release_size=data.frame(starting_N_females_range, success_rates)
write.csv(output_prob_success_by_release_size, "output_prob_success_by_release_size.csv", row.na mes
= F)
```

Conclusions: Based on the results above, at least nine pairs of birds are necessary for release to have an >80% chance of resulting in a stable or increasing population.

For comparison, we assess how release size relates to the probability of avoiding extirpation (i.e., complete failure of release). This is important to consider as even a mildly declining introduced population can still serve as a safe haven for the species while new approaches to handle disease in its home range are developed.

```
set.seed(321) starting_N_females_range
<- 1:30
success_rates <- sapply(starting_N_females_range, function(starting_N_females) { simulate_population(
    pairs_renest_within_season, chicks_per_nest,
    juvenile_survival_mean, juvenile_survival_se,
    adult_survival_mean, adult_survival_se,
    nest_success_upper, nest_success_lower,
    firstyr_juvenile_survival_effect,
    firstyr_adult_survival_effect, firstyr_nest_success_upper,
    firstyr_nest_success_lower,
    starting_N_females, num_years, num_simulations, extirpation_risk=T
  )
})

plot(starting_N_females_range, success_rates, type = "b", pch = 19,
    xlab = "Starting Number of Females", ylab = "Success Rate",  main
    = "Success Rate by Starting Population", ylim=c(0,1))
```

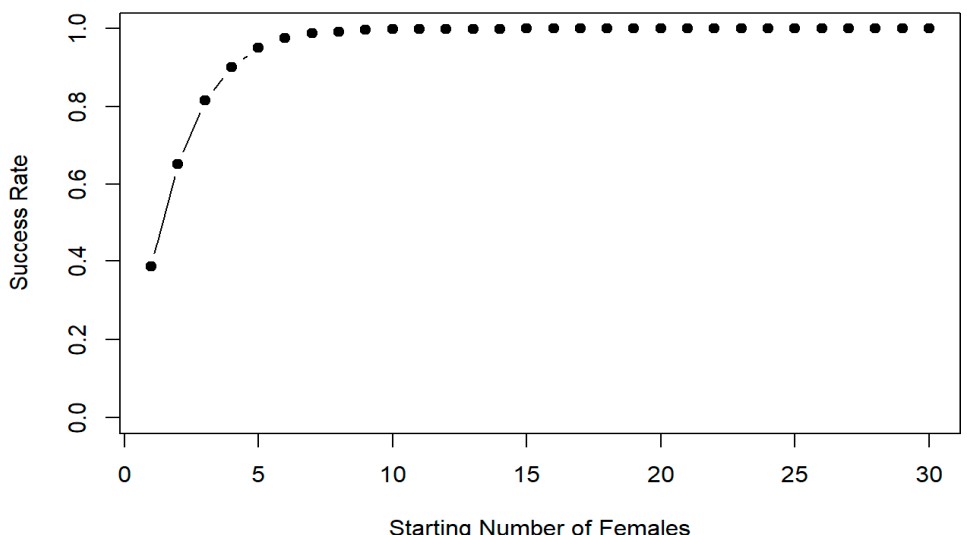

Conclusions: Based on the results above, defining success as the translocated population not being extirpated, at least five pairs of birds are necessary for a release to have a >95% chance of being successful. With eight pairs, that chance rises to >99%.

Based on both results above, we performed multiple sensitivity analyses considering nine pairs and our original goal of population stability/increase.

However, before the sensitivity analyses, we first created PVA estimates for population sizes that spanned the wider range of population sizes possible across the cluster sizes we identified with our lidar-based habitat suitability model (LHSM) analysis, and without considering CI effects on species, so we could estimate the probability of naturalized/established population stability by population size.

```
set.seed(321)
starting_N_females_range <- c(1:30,35, 40, 45, 50, 60, 70, 80, 90, 100)
success_rates <- sapply(starting_N_females_range, function(starting_N_females) {
  simulate_population(
    pairs_renest_within_season, chicks_per_nest,
    juvenile_survival_mean, juvenile_survival_se,
    adult_survival_mean, adult_survival_se,
    nest_success_upper, nest_success_lower,
    firstyr_juvenile_survival_effect=0,
    firstyr_adult_survival_effect=0,
    firstyr_nest_success_upper, firstyr_nest_success_lower,
    starting_N_females, num_years, num_simulations
  )
})

output_prob_stability_by_pop_size=data.frame(starting_N_females_range, success_rates)
names(output_prob_stability_by_pop_size)=c("Population_size", "Prob_of_pop_stability")
write.csv(output_prob_stability_by_pop_size, "output_prob_stability_by_pop_size.csv", row.names
= F)

plot(starting_N_females_range, success_rates, type = "b", pch = 19,
     xlab = "Population size (number of females)", ylab = "Probability of population stability",
     main = "Probability of population stability by size of \n established/naturalized populatio
n", ylim=c(0,1))
```

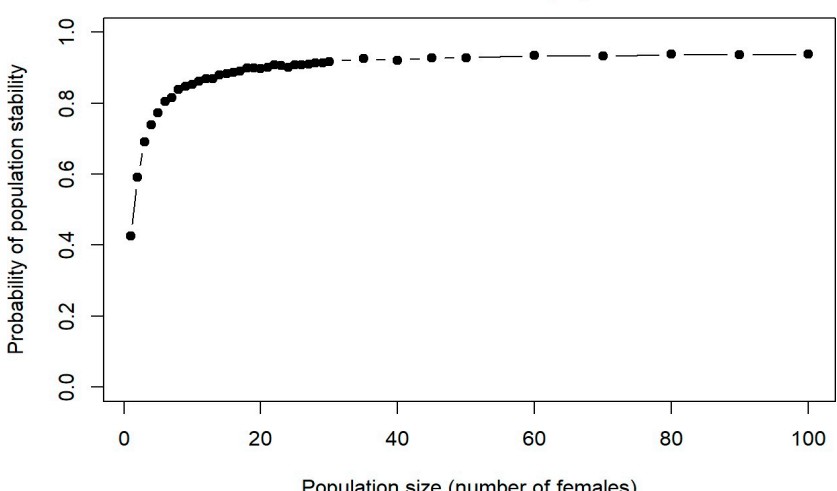

*Appendix B.4. Sensitivity Analyses of PVA*

Survival may vary widely in a new environment; therefore, we performed several sensitivity analyses to explore the effect of varying survival rates on release success. We first tested the effect of CI on long-term adult and juvenile survival.

Appendix B.4.1. CI Success Rate by Long-Term Mean Adult Survival Effect

We ran the PVA under multiple values of long-term mean survival rates of adults (from 0.85 to 1).

```
set.seed(321)
long_term_mean_adult_survival_effect_range <- seq(-0.10, 0.05, 0.01)
success_rates <- sapply(long_term_mean_adult_survival_effect_range, function(long_term_mean_adul
t_survival_effect) {
  adult_survival_mean=adult_survival_mean+long_term_mean_adult_survival_effect
  if (adult_survival_mean>=1) adult_survival_mean = 0.999
  simulate_population(
    pairs_renest_within_season, chicks_per_nest,
    juvenile_survival_mean, juvenile_survival_se,
    adult_survival_mean, adult_survival_se,
    nest_success_upper, nest_success_lower,
    firstyr_juvenile_survival_effect,
    firstyr_adult_survival_effect,
    firstyr_nest_success_upper, firstyr_nest_success_lower,
    starting_N_females, num_years, num_simulations
  )
})

plot(long_term_mean_adult_survival_effect_range+adult_survival_mean, success_rates, type = "b",
pch = 19,
    xlab = "Mean adult survival", ylab = "Success Rate",
    main = "Success Rate by long term mean adult survival", ylim=c(0,1))
abline(v = adult_survival_mean, col = "red", lty = 2)
```

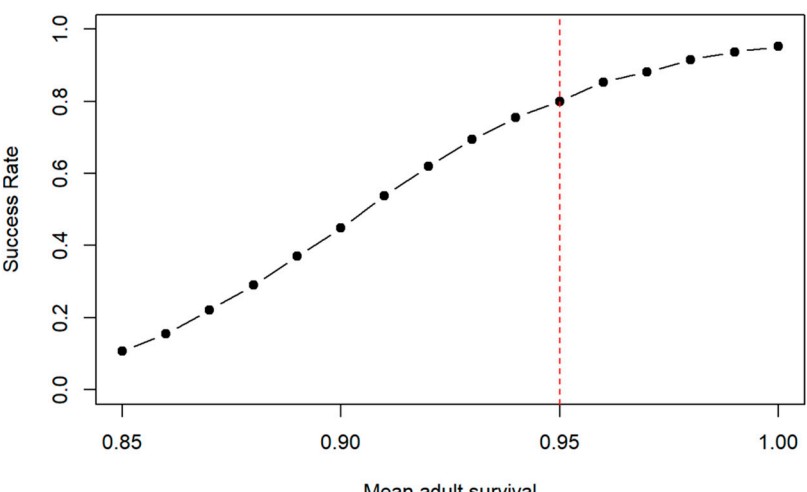

Conclusions: CI success is linearly and positively related to the long-term survival rate of ʻākohekohe in its new environment. Below a 91% adult survival after the initial year post release, the chance of CI success drops below 50%.

Appendix B.4.2. Success Rate by Long-Term Mean Juvenile Survival Effect

We considered values of juvenile survivorship varying from −10 to +5% of the calculated means.

```
set.seed(321)
long_term_mean_juvenile_survival_effect_range <- seq(-0.10, 0.05, 0.01)
success_rates <- sapply(long_term_mean_juvenile_survival_effect_range, function(long_term_mean_j
uvenile_survival_effect) {
  juvenile_survival_mean=juvenile_survival_mean+long_term_mean_juvenile_survival_effect
  if (juvenile_survival_mean>=1) juvenile_survival_mean = 0.999

  simulate_population(
    pairs_renest_within_season, chicks_per_nest,
    juvenile_survival_mean, juvenile_survival_se,
    adult_survival_mean, adult_survival_se,
    nest_success_upper, nest_success_lower,
    firstyr_juvenile_survival_effect,
    firstyr_adult_survival_effect,
    firstyr_nest_success_upper,  firstyr_nest_success_lower,
    starting_N_females, num_years, num_simulations
  )
})

plot(long_term_mean_juvenile_survival_effect_range+juvenile_survival_mean, success_rates, type =
"b", pch = 19,
     xlab = "Mean juvenile survival", ylab = "Success Rate",
     main = "Success rate by long term mean juvenile survival", ylim=c(0,1))
abline(v = juvenile_survival_mean, col = "red", lty = 2)
```

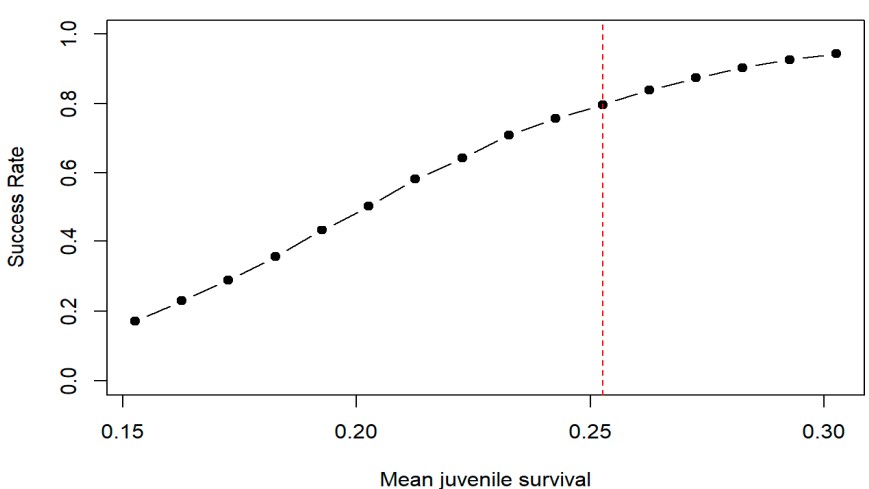

Conclusions: The success of a CI hinges slightly less on juvenile survivorship, in comparison to adult survival, but can still have a large influence on release outcomes if observed values deviate from the estimated mean value across Hawaiian honeycreepers.

Appendix B.4.3. Sensitivity Analysis of CI Success and First-Year Survival Effects on Adults

We calculated the sensitivity of CI success and first year survival effects on adults, considering a wider range of survival values for that first year, given that survival may be largely affected by CI (a −45% to 0% effect on adult survival estimates, translating to mean adult survival ranging from 50% to 95%).

```
set.seed(321)
# starting_N_females=10
first_yr_mean_adult_survival_effect_range <- seq(-0.45, 0, 0.05)
success_rates <- sapply(first_yr_mean_adult_survival_effect_range, function(first_yr_mean_adult_
survival_effect) {
  firstyr_adult_survival_effect=first_yr_mean_adult_survival_effect
  if (adult_survival_mean>=1) adult_survival_mean = 0.999
  simulate_population(
    pairs_renest_within_season, chicks_per_nest,
    juvenile_survival_mean, juvenile_survival_se,
    adult_survival_mean, adult_survival_se,
    nest_success_upper, nest_success_lower,
    firstyr_juvenile_survival_effect,
    firstyr_adult_survival_effect,
    firstyr_nest_success_upper, firstyr_nest_success_lower,
    starting_N_females, num_years, num_simulations
  )
})

plot(first_yr_mean_adult_survival_effect_range+adult_survival_mean, success_rates, type = "b", p
ch = 19,
     xlab = "First year mean adult survival", ylab = "Success Rate",
     main = "Success rate by first year mean adult survival", ylim=c(0,1))
abline(v = adult_survival_mean, col = "red", lty = 2)
```

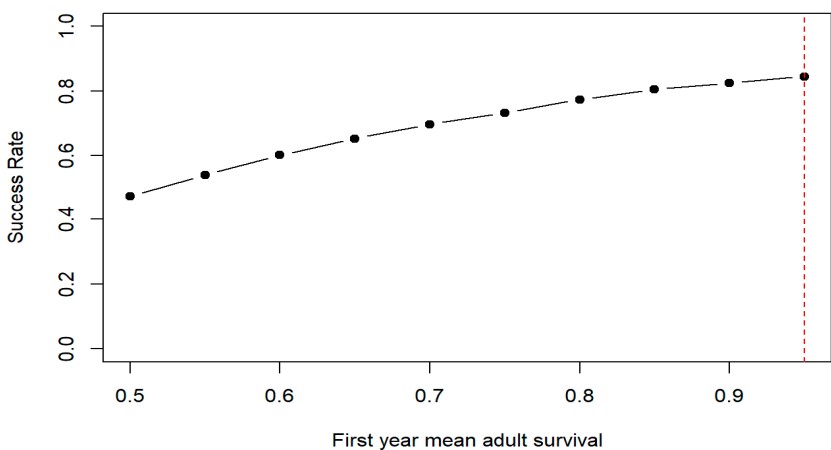

Conclusions: First-year adult survival has a generally small effect on the release success rate; additional runs showed that this effect can be mitigated with larger initial releases. The red dashed line represents the long−term adult survival rate (0.95).

*Appendix B.5. Discussion*

Although the PVA provides a useful initial estimate of release size and 'ākohekohe CI viability to contextualize our estimates of available release habitat, these PVA results warrant careful interpretation given inherent data limitations, and the additional factors detailed below.

Several key demographic parameters, especially juvenile survival, were based on estimates from other Hawaiian honeycreeper species due to the lack of empirical data for 'ākohekohe specifically. Furthermore, the model did not account for potential density dependence effects on vital rates as the population grew. The sensitivity analyses revealed adult survival and first year effects as critical knowledge gaps. The CI success appears to be highly contingent on maintaining high adult survival rates typical of wild 'ākohekohe

populations. If CI substantially reduces long-term adult survival, the population growth may be insufficient to sustain the population, as the models currently predict.

Another factor affecting the viability of releases, which is not incorporated into standard PVAs, is the issue of the dispersal behavior of released birds from release sites. Experiences from similar CIs indicate that introduced birds may attempt to return to their original location or disperse away from the release site, thereby affecting the effective 'survival rate' within the new habitat. This was true for the experimental movement of movement of two other forest bird species, palila (*Loxioides bailleui*) and i'iwi, both species which are less aggressive and territorial than 'ākohekohe [20,58]. Counterbalancing this risk, several homogenous forested areas (HFAs) that our analysis identified as potential CI candidate sites were close to one another which, with some movement by released individuals, could greatly increase the amount of territory available for a new population. We had no data for 'ākohekohe to reduce these uncertainties; thus, this issue of dispersal and movement remains a critical uncertainty that may directly influence the stability and growth of newly established populations.

Another behavior-related limitation of our work is the assumption of the successful pairing of male and female 'ākohekohe post translocation. The variability in pair–bond formation is a pivotal consideration, as our study did not delve into the methods to ensure the required number of breeding pairs, such as translocating known breeding pairs or strategies to encourage the pairing of translocated birds. This represents an area for future research that could be useful for refining the PVAs developed in our study.

Although this first PVA for 'ākohekohe was developed to support wider CI habitat modeling, future work could expand on its utility. For instance, the consideration of supplemental releases could be investigated, as well as a wider set of plausible CI scenarios. The code presented here is a replication of the entire PVA, for those interested in such future analyses.

Lastly, we also performed the analyses above using the same number of individuals but with the goal of avoiding extirpation (a much lower bar); all the sensitivity analyses which were under that goal showed that the likelihood of complete extirpation of the population within that 20-year period was much less sensitive to variability in the survival of both adults and juveniles. This adds extra assurance to our results that, even though the probability of a stable or increasing population under the release considered was not extremely high, the probability we would entirely lose the new population was very low, and not as susceptible to large variability in critical vital rates.

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
