# Peer review of "Identifying Conservation Introduction Sites for Endangered Birds through the Integration of Lidar-Based Habitat Suitability Models and Population Viability Analyses"

_remotesensing, doi:10.3390/rs16040680_

Round 1
Reviewer 1 Report
Comments and Suggestions for Authors

Reviewer 2 Report
Comments and Suggestions for Authors
I have gone through the manuscript with interest. The study brings forward an interesting aspect and analysis of habitat suitability for an important species. After going through the manuscript, I came up with the following comments and suggestions
- the abstract is well written though it is not very necessary to include the tags Background, Methods, Results and Conclusions
- the motivation is clear but it can be further strengthened by expressing the gap in the context of previous studies and bringing out the novelty of the current study
- the quality of Figure 2 and Figure 3 needs to be improved. Axis titles and labels are hardly visible
- in cases such as Figure 2 and Figure A2 where more than one plot is displayed, it is best to also have sub-labels e.g (a) which can then be cited in text. This makes it easy for readers to follow
- the criteria of selection of suitability factors needs to be clearly spelt out and justified. Specifically, the strength of each factor in influencing suitability of the species of interest needs to be well supported by literature
